# Field testing two existing, standardized respiratory severity scores (LIBSS and ReSViNET) in infants presenting with acute respiratory illness to tertiary hospitals in Rwanda – a validation and inter-rater reliability study

**Boniface Hakizimana**[1,2], **Edgar Kalimba**[1,3], **Augustin Ndatinya**[4], **Gemma Saint**[5,6], **Clare van Miert**[7], **Peter Thomas Cartledge**[1,8,9]*

1 Department of Pediatrics, School of Medicine, University of Rwanda, Kigali, Rwanda, 2 Department of Pediatrics, University Teaching Hospital of Kigali, Kigali, Rwanda, 3 King Faisal Hospital, Kigali, Rwanda, 4 Rwanda Military Hospital, Kigali, Rwanda, 5 Institute of Child Health, University of Liverpool, Liverpool, United Kingdom, 6 Department of Respiratory Pediatrics, Alder Hey Children's NHS Foundation Trust, Liverpool, United Kingdom, 7 School of Nursing and Allied Health Liverpool John Moores University, Liverpool, United Kingdom, 8 Department of Emergency Medicine, Yale University, New Haven, Connecticut, United States of America, 9 Rwanda Human Resources for Health (HRH) Program, Ministry of Health, Kigali, Rwanda

* peterthomascartledge@gmail.com

## Abstract

### Introduction

There is a substantial burden of respiratory disease in infants in the sub-Saharan Africa region. Many health care providers (HCPs) that initially receive infants with respiratory distress may not be adequately skilled to differentiate between mild, moderate and severe respiratory symptoms, which may contribute to poor management and outcome. Therefore, respiratory severity scores have the potential to contributing to address this gap.

### Objectives

to field-test the use of two existing standardized bronchiolitis severity scores (LIBSS and ReSViNET) in a population of Rwandan infants (1–12 months) presenting with respiratory illnesses to urban, tertiary, pediatric hospitals and to assess the severity of respiratory distress in these infants and the treatments used.

### Methods

A cross-sectional, validation study, was conducted in four tertiary hospitals in Rwanda. Infants presenting with difficulty in breathing were included. The LIBSS and ReSViNET scores were independently employed by nurses and residents to assess the severity of disease in each infant.

**Data Availability Statement:** Availability of data and materials: The study data-set is available online at https://doi.org/10.7910/DVN/N4O05G/AKSNOZ.

**Funding:** The author(s) received no specific funding for this work.

**Competing interests:** The authors have declared that no competing interests exist.

**Abbreviations:** aROC, Area under Receiver Operating Characteristic; BUTH, Butare University Teaching Hospital; CPAP, Continuous Positive Airway Pressure; CLD, Chronic Lung Disease; ER, Emergency Room; HCPs, Health Care Providers; HIC, High-Income Countries; HDU, High Dependency Unit; IMCI, Integrated Management of Childhood Illness; LIC, Low-Income Country; KFH, King Faisal Hospital; KUTH, Kigali University Teaching Hospital; LIBSS, Liverpool Infant Bronchiolitis Severity Score; OPD, Outpatient Department; ORS, Oral Rehydration Salt; PI, Principal Investigator; PICU, Pediatric Intensive Care Unit; ReSViNET, Respiratory Syncytial Virus Network Scale; RMH, Rwanda Military Hospital; WHO, World Health Organization.

## Results

100 infants were recruited with a mean age of seven months. Infants presented with pneumonia (n = 51), bronchiolitis (n = 36) and other infectious respiratory illnesses (n = 13). Thirty-three infants had severe disease and survival was 94% using nurse applied LIBSS. Regarding inter-rater reliability, the intra-class correlation coefficient (ICC) for LIBSS and ReSViNET between nurses and residents was 0.985 (95% CI: 0.98–0.99) and 0.980 (0.97–0.99). The convergent validity (Pearson's correlation) between LIBSS and ReSViNET for nurses and residents was R = 0.836 (p<0.001) and R = 0.815 (p<0.001). The area under the Receiver Operator Curve (aROC) for admission to PICU or HDU was 0.956 (CI: 0.92–0.99, p<0.001) and 0.880 (CI: 0.80–0.96, p<0.001) for nurse completed LIBSS and ReSViNET respectively.

## Conclusion

LIBSS and ReSViNET were designed for infants with bronchiolitis in resource-rich settings. Both LIBSS and ReSViNET demonstrated good reliability and validity results, in this cohort of patients presenting to tertiary level hospitals. This early data demonstrate that these two scores have the potential to be used in conjunction with clinical reasoning to identify infants at increased risk of clinical deterioration and allow timely admission, treatment escalation and therefore support resource allocation in Rwanda.

## Introduction

Acute respiratory distress is a frequent cause of pediatric emergency department attendance. Common etiologies include bronchiolitis and pneumonia [1, 2]. The burden of respiratory disease remains in Low-Income Countries (LICs), where mechanical ventilation facilities are limited or unavailable [3]. Deaths from pneumonia and bronchiolitis have been linked to low health coverage, lack of exclusive breastfeeding, malnutrition, incomplete immunization and lack of access to an appropriate health care service [4, 5]. Many health care providers (HCPs) that initially receive infants with respiratory distress may not be adequately skilled to differentiate between mild, moderate and severe respiratory symptoms, which may contribute to poor management and outcome [6]. Untreated or unsupported pediatric respiratory distress can lead to respiratory failure, identified as the main cause of cardiac arrest in children [7]. Continuous positive airway pressure (CPAP) can help to prevent progression to respiratory failure [3]. However, identifying those children who need admission and those that need higher levels of care is dependent upon HCPs with the necessary skills and tools.

Risk prediction models offer the potential to support such clinical decision making, and in respiratory disease can help to identify children that require admission and respiratory support [8]. One such model is a clinical measurement instrument consisting of clinical signs and symptoms that are grouped together to measure an intended construct. The validation of such an instrument should ideally include: assessment of reliability, responsiveness, and usability [9–15]. Instruments could be used by HCPs in their decision making, such as when to admit children, or when to escalate care of infants with respiratory distress. In order to be fit for purpose, such an instrument should be validated, reliable, quick to perform, and straightforward to interpret in the clinical context [15]. It should not involve complex measurements, descriptions or equipment.

There are several respiratory distress instruments from high-income countries (HIC), with many being developed specifically for single pathologies such as bronchiolitis [9, 15–33]. Several models have been developed in resource-limited settings, to assess for primary outcomes of mortality or antibiotic-treatment failure in children with severe bacterial pneumonia, namely; The Mamtani score from India, modified Respiratory Index of Severity in Children (RISC) from Kenya, RISC-Malawi and the original RISC Score from South Africa [15, 34–38]. However, the income and health provisions in these countries are broad, varying between low-income (LIC) Malawi, and upper-middle income (UMIC) South Africa.

The Liverpool Bronchiolitis Severity Score (LIBSS) and the Respiratory Syncytial Virus Network Scale (ReSViNET), are both bronchiolitis specific scores. LIBSS and ReSViNET have both previously been assessed for apparent validity (to develop the model) and internal validity in infants in developed countries with bronchiolitis symptoms [25, 28].

## Objectives

This study sought to field-test the use of two scoring instruments (LIBSS and ReSViNET), assessing the severity of respiratory distress in a population of Rwandan infants (1–12 months) presenting to urban, tertiary, pediatric hospitals. Specifically, we assessed the usability, inter-rater reliability and internal consistency of the two instruments. The secondary aim was the description of the severity of respiratory distress in these infants and the treatments used.

## Reasoning for choice of LIBSS and ReSViNET scores

Both the LIBSS and ReSViNET instruments share five parameters and employ clinical parameters assessed by HCPs [15]. The parameters of both scores are applicable to all respiratory diseases that can cause respiratory distress in infants in settings such as Rwanda, and can be undertaken with no equipment (ReSViNET) or with just a saturation monitor (LIBSS). Specifically we sought to continue the work of the LIBSS and ReSViNET teams to see if these scores could be used in our setting, without requiring an assessment of HIV, malaria or chronic nutritional status, which are required for RISC scores.

# Materials and methods

## Study design

This was a cross-sectional, multi-centre, validation study, which took place from September 2018 to February 2019. Reporting of this study has been verified in accordance with the TRIPOD checklist for reporting prediction models [39, 40].

## Study sites (data source)

The principal investigator (BH) undertook the study to complete his MMed pediatric residency at the University of Rwanda. The study was therefore conducted at four public university, or university affiliated hospitals in Rwanda, namely; Kigali University Teaching hospital (CHUK), Rwanda Military Hospital (RMH), Butare University Teaching hospital (CHUB) and Ruhengeri Referral Hospital (RRH). Recruitment was planned but unsuccessful at King Faisal Hospital (KFH), a collaborative public-private, university, tertiary hospital. CHUK, KFH and RMH are all located in Kigali city, the capital of Rwanda. CHUB and RRH are in provincial towns. All hospitals are tertiary referral centres. These sites are all located in urban settings but receive patients from both rural and urban settings. Patients using the national health insurance system ("mutuelle de sante") cannot self-refer to tertiary hospitals and are therefore

referred from Health Centers and District Hospitals. A small number of paying patients will self-refer.

## Study population

Participants were recruited prospectively from children in both outpatient departments and the pediatric emergency rooms (ER) at the study sites.

**Inclusion criteria.** infants 1–12 months of age presenting with respiratory distress due to any respiratory illness and whose parents could give consent [41, 42].

Our case definition of symptoms and signs indicative of respiratory distress were: apnea, subcostal or intercostal recession, tracheal tug, nasal flaring, head bobbing, grunting, cyanosis, oxygen desaturation, tachypnea, wheezing, stridor, oxygen requirement, or reduced air-entry [25, 28].

**Exclusion criteria.** infants with known chronic lung disease (CLD), or infants who presented with a non-respiratory cause of respiratory distress (e.g. cardiac disease).

**Sampling.** recruitment was opportunistic at the study sites, with infants presenting to the study sites and meeting the case definition of respiratory distress being approached for recruitment.

**Patient recruitment.** The resident pediatrician and/or resident on duty identified eligible infants. Parents were provided with both verbal and written information about the study. If they agreed to participate in the study, written informed consent was gained and demographic details collected.

## Clinical care of infants

All data-collectors were HCPs and were involved in the clinical care of the participants, and so were not blind to the infants' condition, treatment interventions and patient outcomes. If an infant required stabilization this was done before recruitment and severity scoring. These interventions were administration of oxygen, antibiotics and fluids.

## Data collection tool

Five data collection tools were used (https://doi.org/10.7910/DVN/N4O05G):

1. Unique Patient Identifier Sheet

2. Study-specific questionnaire: Patient demographics guided by the Demographics and Health Survey (DHS) parameters [41].

3. LIBSS: A validated score for use in children with bronchiolitis between the ages of 0–12 months. LIBSS has two scoring systems 0–3 months score and 3–12 months to take into account for age-dependent vital signs [28]. The LIBSS includes ten parameters, namely; General condition; Apnea; Increased work of breathing; Sa02; Respiratory Rate; Appearance; Heart Rate; Feeding; Urine output; Capillary refill time.

4. ReSViNET: a validated clinical severity scale that was developed to assess the severity of illness in infants with bronchiolitis [25, 43]. ReSViNET includes seven parameters, namely; Feeding intolerance; Medical intervention; Respiratory difficulty; Respiratory rate; Apnea; General condition; Fever

5. Study-specific follow-up questionnaire for final outcomes: admission to pediatric ward; admission to high dependency unit (HDU) or pediatric intensive care unit (PICU); or

outpatient treatment. This section of the questionnaire was specifically designed for the purposes of this study.

### Instrument translation

Most HCPs in Rwanda speak Kinyarwanda (local language), English and/or French. However, English may not be the first language. Therefore, to overcome language-related barriers, LIBSS and ReSViNET were translated into Kinyarwanda by the principal investigator (PI) and then back-translated for accuracy by a native-Kinyarwanda speaking pediatric resident. Translation discrepancies were reviewed with a third native-Kinyarwanda speaking pediatric resident for consensus. Within the scoring instruments, the Kinyarwanda translation was presented alongside the original English.

**Instrument availability.** The Translated LIBSS and ReSViNET, used in this study, are available online (https://doi.org/10.7910/DVN/N4O05G). The original ReSVinet was published in PLoS and is therefore available under Creative Commons licence CC BY [25] and the original LIBSS is available under a CC BY attribution [28].

### Data-collection

One pediatric resident and one nurse independently assessed each infant using LIBSS and ReSViNET. The resident and nurse were instructed to not share their LIBSS/ReSViNET findings with each other. Infants were recruited and scored within an hour of presentation to hospital.

### Training data-collectors/assessors

Prior to undertaking assessments, HCPs were given three hours of training on two consecutive days on how to use the two scores and questionnaires.

### Sample size

Data was collected on paper forms, entered into Microsoft Excel and analyzed using SPSS version 24. The 'rule of thumb' to determine the sample size for a clinical field-test is 10–15 times the number of parameters in a test [44]. This gave a minimum sample size of 100 for LIBSS (10 parameters) and 70 for ReSViNET (7 parameters).

### Outcomes, data management and statistical methods

**Severity of disease.** As there is no "gold standard" for determining the severity of respiratory distress, clinical severity was defined by outpatient management (mild), admission (moderate) and severe (HDU/PICI admission). We determined the severity of illness using the LIBSS and ReSViNET pre-defined cut-points of Mild (0–10); Moderate (11–20); Severe (>21) for LIBSS [28] and 0–6 for mild affection, 7–13 for moderate distress, and 14–20 using ReSViNET [25, 43].

**Validity.** Criterion and convergent validity were used to assess whether the instrument measures what it was intended to measure. Criterion validity (predictive) testing was determined using the area under aROC based on severity scores of nurse-assessed LIBSS and ReSViNET. aROCs of 0.50–0.70, 0.70–0.90 and >0.90 were pre-defined as indicating low, moderate and high validity respectively [44]. Construct validity was assessed by examining the correlation between LIBSS and other measures of the impact of the disease: hospital admission, length of stay, HDU/PICU admission and mortality. Length of stay was categorised into binary criteria of long/short dependent on the median length of stay as a cut-off.

*The two scores use 12 different parameters. Five parameters are shared by the two tools, namely*: *Apnea; Feeding intolerance; General condition; Increased work of breathing; and respiratory rate. ReSViNET uses two additional parameters (Fever and medical intervention), and LIBSS employs five unique parameters (Appearance, Capillary refil time, heart rate, oxygen requirement and urine output). Therefore*, convergent validity of LIBSS and ReSViNET scores was assessed using Pearson's r correlation coefficient. Coefficients of 0.4–0.59, 0.6–0.79 and 0.8–1.0 indicating moderate, strong and very strong relationships respectively [45].

**Reliability testing.** Inter-rater reliability (agreement) was evaluated between paired pediatric resident and nurse responses using the Intraclass Correlation Co-efficiency (ICC) with ICC <0.75 poor to moderate, >0.75 is good, >0.9 is excellent [44, 46]. Internal consistency of the parameters within each score was measured using Cronbach alpha, whereby Cronbach's: <0.70 poor, >0.70 good (if <7 items), interpretation is dependent on number of parameters [44].

## Ethical considerations and declarations

**Financial disclosure.** The author(s) sought nor received any specific funding for this work.

**Potential conflict of interest.** The authors have declared that no competing interests exist.

**Confidentiality.** Unique Individual Patient Identifier codes were used when completing the questionnaires. Patient-identifiable information was kept in a separate database to protect confidentiality.

**Informed consent.** Informed, written consent, was gained from parents or caregivers.

**Incentives for subjects.** No incentives, financial or otherwise, were offered to participants.

**Risk to subjects (including safeguards to mitigate these risks).** No significant physical, social, emotional, legal or financial risks were identified.

**Ethical approval.** The project proposal was reviewed by the Institutional Review Board (IRB) at the University of Rwanda. Ethical approval was given (Ref: NO237/CMHSIRB/2018). The proposal was then reviewed and approved by the ethics committees of each enrolling site.

*Availability of data and materials*: The study data-set is available online at https://doi.org/10.7910/DVN/N4O05G/AKSNOZ.

## Results

### Recruitment

A total of n = 107 eligible infants were recruited (Fig 1): CHUK (n = 47); RRH (n = 46), CHUB (n = 9) and RMH (n = 5). Data from seven infants could not be used due to data collection being incomplete or inadequate, and therefore, these cases were removed. Data most commonly omitted were the heart rate and respiratory rate.

### Data quality

No data points were missing from the final data set.

### Demographic characteristics of the participants

Mean age of participants was seven months (201 days, Standard deviation, SD ±114.7), and 63% were male. Mean weight and median length of hospital stay (LoS) were 6.6kg (SD±2.4) and 4.0 days (min = 0, max = 39) respectively (Table 1). Participants were diagnosed with pneumonia (n = 51, 51%), bronchiolitis (n = 36, 36%), viral induced wheezing (n = 6, 6%), upper respiratory

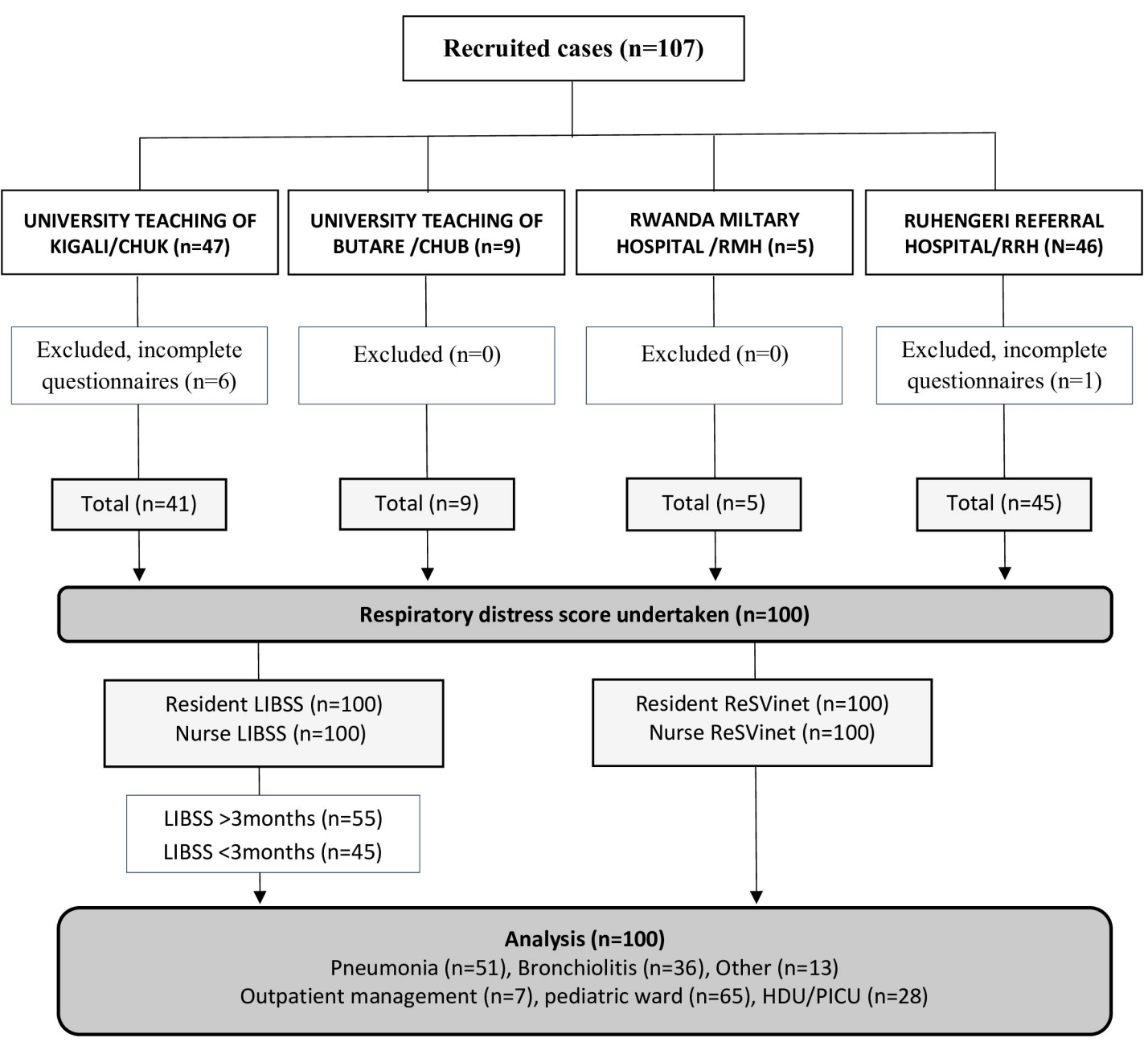

**Fig 1. CONSORT diagram.**

tract infection (n = 1, 1%) and other infectious respiratory illnesses (n = 6, 6%). Survival rate was 94% (n = 94). Twenty-eight patients required HDU or PICU management, with 7% (n = 7) requiring outpatient management alone. The most common co-morbidities were: malnutrition (n = 12, 12%), prematurity (n = 5, 5%) and HIV (n = 2, 2%) (Table 1).

## Severity of disease

ReSViNET described fewer infants (n = 16, 16%) as having severe disease compared to LIBSS (n = 33, 33%) (Table 2). Most patients presented in moderate or severe respiratory

**Table 1. Demographic characteristics of the participants.**

| Characteristic | |
|---|---|
| **Gender** | |
| **Male** | 63 (63%) |
| **Female** | 37 (37%) |
| **Mean age (days)** | 201 (SD ±114.73) |
| **Median** age **(days)** | 204 |
| **Mean weight (kg)** | 6.6 (SD ± 2.43) |
| **Median weight (kg)** | 6.0 |
| **Social Group (Ubedehe[U])** | |
| **High (3 & 4)** | 47 (47%) |
| **Low (1 & 2)** | 53 (53%) |
| **Residence** | |
| **Urban** | 32 (32%) |
| **Rural** | 68 (68%) |
| **Living sibling** | |
| **No** | 20 (20%) |
| **Yes** | 80 (80%) |
| **Vaccinations complete** | |
| **No** | 0 (0%) |
| **Yes** | 100 (100%) |
| **Maternal marital status** | |
| **Married** | 84 (84%) |
| **Single, divorced, Widowed** | 16 (16%) |
| **Maternal age** | |
| **Young (<25 years)** | 32 (32%) |
| **Old** (≥25years) | 68 (68%) |
| **Maternal occupation** | |
| **Unemployed** | 19 (19%) |
| **Manual labourer** | 64 (64%) |
| **Professional** | 17 (17%) |
| **Maternal educational level[E]** | |
| **High** | 49 (49%) |
| **Low** | 51 (51%) |
| **Final diagnosis** | |
| **Pneumonia** | 51 (51%) |
| **Bronchiolitis** | 36 (36%) |
| **Viral induced wheezing** | 6 (6%) |
| **Upper respiratory tract infection** | 1 (1%) |
| Other | 6 (6%) |
| **Co-morbidities** | |
| **Malnutrition** | 12 (12%) |
| **Prematurity** | 5 (5%) |
| **HIV positive** | 2 (2%) |
| **Management** | |
| **Outpatient** | 7 (7%) |
| **Pediatric ward** | 65 (65%) |
| **HDU/PICU** | 28 (28%) |
| **Median length of stay (days)** | 4.0 |

(*Continued*)

**Table 1.** (Continued)

| Characteristic | |
|---|---|
| **Mean length of stay (days)** | 6.5 (SD ±7.14) |
| **Mortality rate/Died** | |
| **Yes** | 6 (6%) |
| **No** | 94 (94%) |

[U]Ubedehe is the Rwandan community based social classification;

[E] High = secondary or university completed, Low = primary or no formal education.

distress with 87% (n = 87) and 84% (n = 84) having moderate/severe disease on LIBSS and ReSViNET respectively.

## Validity

**Convergent validity.** The Pearson's correlation between ReSViNET and LIBSS for residents (R = 0.815) and nurses (R = 0.836) were both very strong (Table 3 and Fig 2) [45].

**Criterion validity (predictive).** Both LIBSS and ReSViNET performed well for predicting hospital admission, HDU/PICU admission and mortality (Table 4). However, they performed only moderately well for predicting prolonged length of stay (Fig 3).

## Reliability

**Interrater reliability.** The inter-rater reliability between residents and nurses was excellent for both LIBSS (ICC = 0.985) and ReSViNET (ICC = 0.980) (Table 3).

**Internal reliability (consistency).** Internal consistency for both scores was good with marginally higher internal consistency for data from the ReSViNET score (Cronbach = 0.850 and 0.848 for nurse and resident scoring respectively) (Table 4). Table 4 demonstrates the reliability of each score if each item is deleted from the score, with only the removal of apnea resulting a modest increased reliability in both scores.

## Treatment

Most participants were treated with antibiotics (pneumonia 100%, bronchiolitis 72% and other infections 39%) and oxygen therapy (93%) (Table 5). In infants with bronchiolitis, non-standard therapy including adrenaline (47%), salbutamol nebulization (58%) and steroids (14%). In patients with severe disease, 5% required intubation and mechanical ventilation while 8% required CPAP (Table 5).

## Discussion

This study sought to field-test the use of two scoring instruments (LIBSS and ReSViNET), assessing the severity of respiratory distress in a population of 100 Rwandan infants (1–12

**Table 2. Severity of distress LIBSS and ReSViNET.**

| | | Mild | Moderate | Severe |
|---|---|---|---|---|
| **LIBSS** | Residents | 14 | 48 | 38 |
| | Nurses | 13 | 54 | 33 |
| **ReSVinet** | Residents | 16 | 67 | 17 |
| | Nurses | 16 | 68 | 16 |

**Table 3. Validity and reliability results.**

| | | LIBSS | ReSVinet |
|---|---|---|---|
| Validity statistics | | | |
| **Convergent validity (LIBSS versus ReSViNET)** | Pearson's correlation (resident) | R = 0.815 (CI: 0.70–0.93) (p<0.001) | |
| | Pearson's correlation (nurse) | R = 0.836 (CI: 0.73–0.95) (p<0.001) | |
| **Criterion Validity for Hospital admission** | aROC (nurse) | 0.956 (CI: 0.88–1.0) (p<0.001) | 0.973 (CI: 0.94–1.0) (p<0.001) |
| | aROC (resident) | 0.955 (CI: 0.87–1.00) (p<0.001) | 0.956 (CI: 0.92–0.99) (p<0.001) |
| **Criterion Validity for HDU/PICU** | aROC (nurse) | 0.956 (CI: 0.92–0.99) (p<0.001) | 0.880 (CI: 0.80–0.96) (p<0.001) |
| | aROC (resident) | 0.951 (CI: 0.91–0.99) (p<0.001) | 0.872 (CI: 0.787–0.957) (p<0.001) |
| **Criterion Validity for mortality** | aROC (nurse) | 0.976 (CI: 0.95–1.0) (p<0.001) | 0.974 (CI: 0.944–1.0) (p<0.001) |
| | aROC (resident) | 0.974 (CI: 0.94–1.0) (p<0.001) | 0.980 (CI: 0.954–1.0) (p<0.001) |
| **Criterion Validity for Length of hospital stay** | aROC (nurse) | 0.718 (CI: 0.62–0.82) (p<0.001) | 0.637 (CI: 0.531–0.747) (p<0.001) |
| | aROC (resident) | 0.722 (CI: 0.62–0. 82) (p<0.001) | 0.639 (CI: 0.531–0.747) (p<0.001) |
| **Reliability statistics** | | | |
| **Inter-rater reliability** | Intra-class correlation (Nurse to resident) | 0.985 (CI: 0.98–0.99) (SD±16.741) (p<0.001) | 0.980 (CI: 0.97–0.99) (SD±6.899) (p<0.001) |

HDU = High dependency unit; PICU = Pediatric Intensive Care Unit.

Suggested Interpretation of Validity and Reliability statistics.

aROC (area under Receiver Operating Characteristic): 0.50 = no different than random (i.e. useless), 0.50–0.70 low; 0.70–0.90 moderate, >0.90 high [44].

Intra-class correlation (ICC): <0.75 poor to moderate, >0.75 is good, >0.9 is excellent [44, 46].

Pearson R correlation: R = 0–0.19 very weak, R = 0.2–0.39 weak, R = 0.40–0.59 moderate, R = 0.6–0.79 strong and R = 0.8–1 very strong correlation [45].

months) consulting urban, tertiary, pediatric hospitals. The predictive validity, reliability between raters, and the internal consistency of the two instruments was measured and both instruments performed well.

## Severity of disease

The majority of patients had moderate or severe disease. This likely represents the location of the field testing and the health structure in Rwanda. Children cannot self-present to tertiary sites and are therefore referred from Health Centers and District Hospitals, where they will likely have provided the necessary care of infants with mild disease, without referring to the tertiary hospitals. The pre-established cut-points of ReSViNET identified fewer infants as having severe disease.

## Treatments used

Though assessing severity is important, this study also highlights the importance of the implementation of evidence into clinical practice to ensure that evidence-based treatments are employed. Many patients received unnecessary treatments (Table 5). Antibiotics, adrenaline and/or bronchodilators or not warranted in bronchiolitis but were used frequently in children with this condition and discontinuing the use of these three interventions as a key priority [47–49]. Not only is the efficacy of these medications not backed up by the evidence in the literature, they are also costly for the health care system and families. Further work needs to be undertaken into how to reduce the use of these ineffective treatments. This is not a Rwandan-specific issue, and this problem has also been described in developed [50].

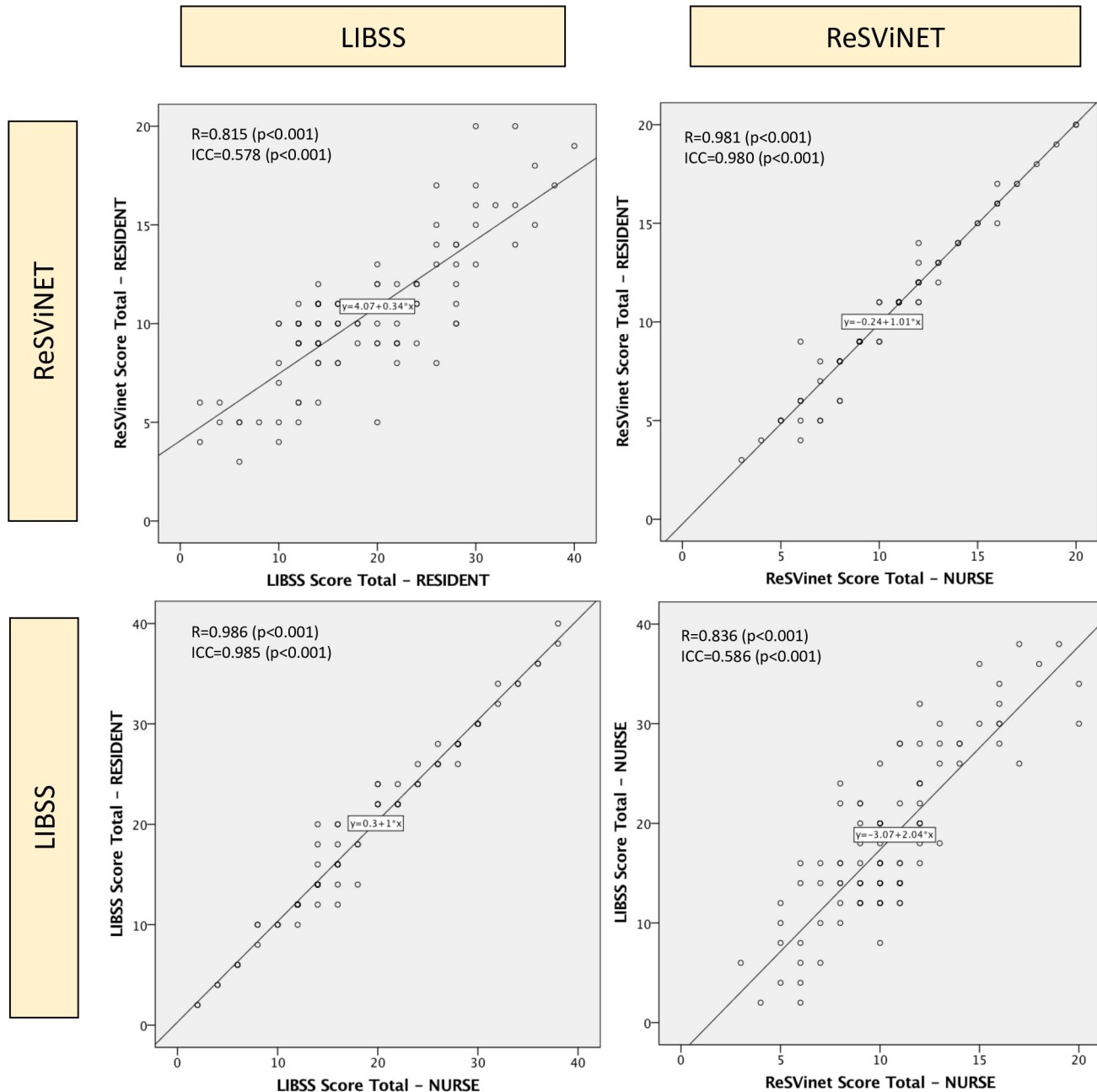

**Fig 2. Convergent validity.** footnote: R = Pearson's coefficient, ICC = Intra-class correlation.

## Validity of the scores

Many healthcare facilities in resource-limited settings will not have medical doctors. There-fore, the nurse-assessed LIBSS and ReSViNET were used for the validity analyses. There is no "gold-standard" for assessing respiratory disease. When LIBSS and ReSViNET were measured against each other, there was a strong correlation between the two scores (Pearson's >0.8). The scores use 12 parameters, and as they share five of parameters (apnoea, feeding intoler-ance, general condition and respiratory rate), the strong correlation is perhaps not surprising, but it is reassuring that they are scoring severity similarly. Both scores performed highly for

**Table 4. Internal reliability (internal consistency) of LIBSS & ReSViNET.**

| | Parameters | LIBSS | | ReSVinet | |
|---|---|---|---|---|---|
| | | NURSE | RESIDENTS | NURSE | RESIDENTS |
| **Overall Scale Cronbach** | | **0.831** | **0.823** | **0.850** | **0.848** |
| **Reliability if item deleted from scale** | | | | | |
| LIBSS only | Appearance | 0.804 | 0.799 | - | - |
| | Central capillary refill time | 0.820 | 0.810 | - | - |
| | Heart rate | 0.826 | 0.822 | | |
| | Oxygen requirement | 0.809 | 0.796 | - | - |
| | Urine output | 0.804 | 0.801 | - | - |
| Shared parameters | Apnea | 0.833 | 0.824 | 0.870 | 0.864 |
| | Feeding | 0.804 | 0.789 | 0.810 | 0.817 |
| | General condition | 0.810 | 0.808 | 0.823 | 0.812 |
| | Increased work of breathing | 0.809 | 0.801 | 0.811 | 0.808 |
| | Respiratory rate | 0.831 | 0.821 | 0.812 | 0.810 |
| ReSVinet only | Fever | - | - | 0.844 | 0.843 |
| | Medical intervention | - | - | 0.829 | 0.825 |

**Suggested Interpretation of Validity and Reliability statistics:** Cronbach's: <0.70 poor, >0.70 good (if <7 items), interpretation is dependent on number of parameters [44].

predicting hospital admission, HDU/PICU admission and mortality. HDU/ICU are not infrequently needed to optimize respiratory and medical support, especially in sub-Saharan Africa [2], however, such care may require the transfer of the patient, is expensive and labour-intensive. Identifying the right patients for this level of care is, therefore, important. The case-definition used in our inclusion criteria overlapped with the scoring systems that were measured. Therefore, when interpreting the validity of the instruments it is important to consider that the sample taken were done so using this case-definition and therefore it is feasible that the tools will perform very differently in unselected children and in primary care levels.

## Reliability

**Interrater reliability.** The inter-rater reliability was good, with lower confidence limits of 0.98 and 0.97 for LIBSS and ReSViNET, whereby scoring >0.9 is "excellent" [44, 46]. In the UK, field-testing of LIBSS, the lowest confidence interval for ICC was 0.75 [28], and the ReSViNET gained lowest ICCs of 0.76 between professionals [25]. It is therefore interesting that the Rwandan professionals scored considerably higher than both these original settings.

**Internal consistency.** The consistency was better within the ReSViNET, and this is despite it having fewer parameters (seven versus ten). It is well known that some items may particularly affect reliability within the data and removing this item could see an improvement in internal consistency, at the cost of content validity. Our data revealed good internal reliability and only the removal of the measurement of apnoea would have marginally improved the internal reliability. As Apnoea is an important aspect of respiratory illness in infants (content validity) we would not advocate removing it.

## Use of LIBSS and ReSViNET in the Rwandan setting

The most common co-morbidities were: malnutrition (12%), prematurity (5%) and HIV (2%) (Table 2). According to the Rwandan DHS 2015 data, 38% of under-5 children are stunted where 18% of infants between 6 to 8 months are stunted [51]. RISC requires and assessment of

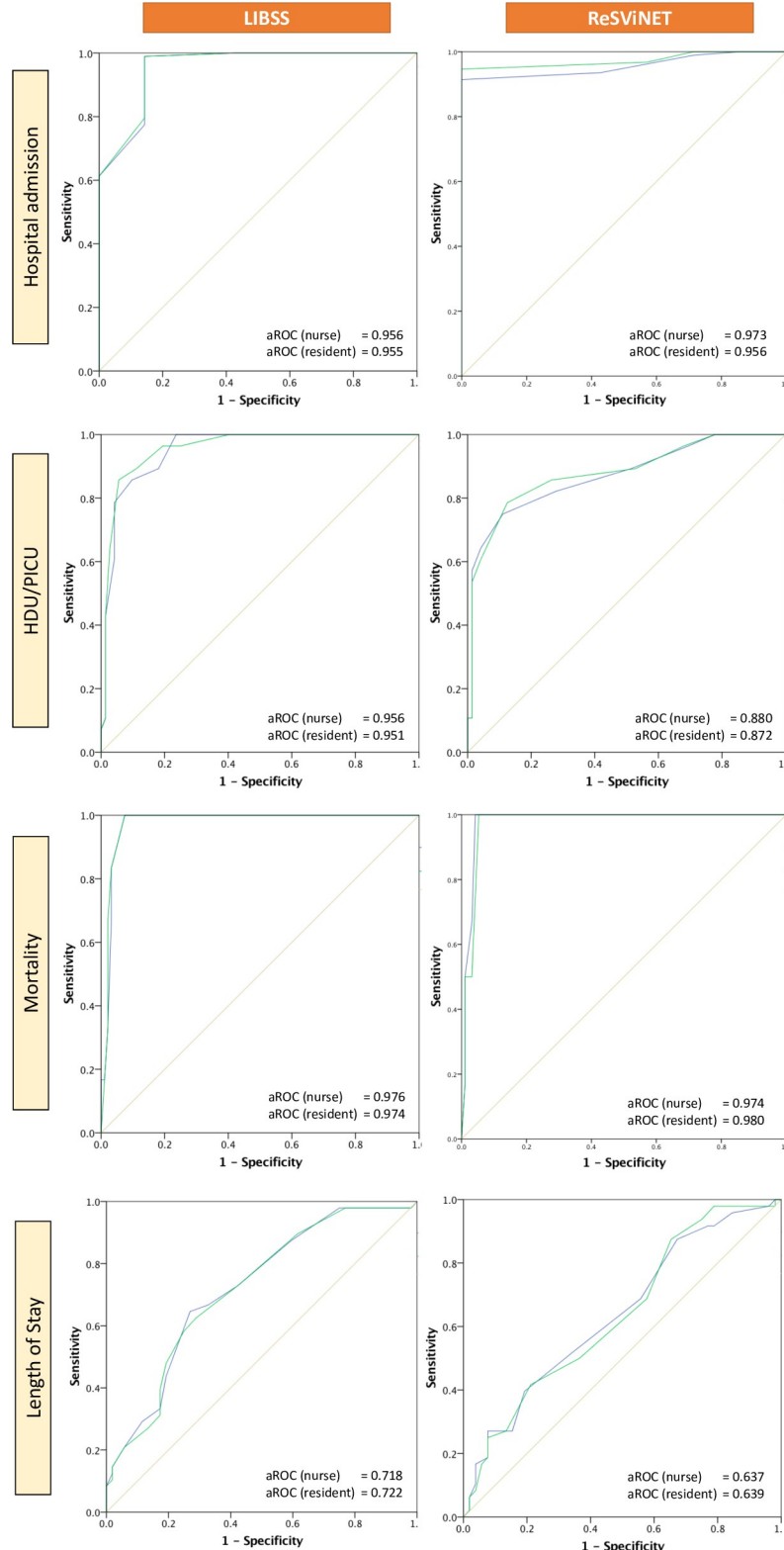

**Fig 3. Criterion validity (aROC) of nurse (green) and resident (blue) performed LIBSS and ReSVinet.** footnote: aROC = area under Receiver Operating Characteristic.

**Table 5. Treatments used.**

| | Pneumonia (n = 51) | Bronchiolitis (n = 36) | Other (n = 13) |
|---|---|---|---|
| **Antibiotic** | 51 (100%) | 26 (72.2%) | 5 (38.5%) |
| **Oxygen therapy** | 50 (98%) | 35 (97.2%) | 8 (61.5%) |
| **Salbutamol nebulisation** | 13 (25.5%) | 21 (58.3%) | 8 (61.5%) |
| **CPAP** (*continuous positive airway pressure*) | 5 (9.8%) | 3 (8.3%) | 0 (0.0%) |
| **Adrenaline nebulization** | 4 (7.8%) | 17 (47%) | 4 (30.8%) |
| **Steroid administration** | 3 (5.9%) | 5 (13.9%) | 4 (30.8%) |
| **Intubation and ventilation** | 1 (2%) | 3 (8.3%) | 1 (7.7%) |

nutritional status as these infants are particularly prone to worse outcomes. Therefore, using a respiratory distress score including HIV and/or nutritional assessment is likely to be beneficial. However, doing so requires additional skills (nutritional assessment) which can add to the complexity of the scoring.

## Comparison of the population with original tools

The major consideration here is that the original scoring tools were designed for infants from resource-rich settings and for infants with bronchiolitis alone, with other respiratory illnesses being assessed. The cut-points for disease severity were therefore determined in a different setting with cohorts of patients very different from ours. These factors may account for the lower criterion validity for HDU/PICU admission with ReSViNET (aROC 0.880) compared to LIBSS (0.956).

## Use of valid and reliable clinical scores in research

As well as using scores in clinical practice severity scores are also useful for clinical trials, in order to consistently and reliably assess severity of diseases and the effects of treatments. Both the scores described here are simple to use, without complex investigations or assessments, and have the potential to be used in research activities in this setting and age group.

## Bias

Our population of outpatient-managed cases was only small (n = 7), probably reflecting the type of hospitals where assessment was undertaken. Usually tertiary and/or referral hospitals in Rwanda receive critically ill patients transferred from peripheral hospitals. Due to the scarcity of hospital beds, priority is given to the most critically ill patients. But self-referred patients with private financial capacity can consult as outpatients. Therefore our population was biased to either tertiary, referred patients or private self-referring patients.

## Limitations

Limitations of our study include the lack of a gold standard for the evaluation of children with acute respiratory distress and the lack of a mechanism to have an ongoing follow-up. The other major limitation was our inability to hire full-time data collector nurses at each study site. This would have potentially increased recruitment and reduced incomplete datasets that led to seven cases being excluded. We attempted to recruit at a University public-private hospital, King Faisal Hospital (KFH), a public-private, but only recruited three cases, which was felt insufficient for inclusion. The reason for poor recruitment was high perceived work load and reported low admissions of patients with respiratory distress.

The sample size was suitable for field-testing the severity scores and to assess if they are feasible for use. There is however a risk of over-fitting in relation to the prediction measures (aROCs). The tests we used were created for assessing bronchiolitis rather than all causes of respiratory illness. They do not assess for HIV, malaria or chronic nutritional status, and these may have important prognostic implications that require additional resources and HCP skills, therefore, may impede utilization both in clinical and research applications.

## Next steps

In this study, it was not feasible to undertake repeated assessments, and therefore, responsiveness was not assessed. Testing responsiveness could have allowed us to monitor progress in hospital and see if the scores were helpful to identify deterioration or improvement and guide care. Applying evidence into practice is challenging. These two scores have not been applied in Rwanda and doing so would require additional work, such as assessing stakeholder attitudes, along with addressing training needs of those using scores in primary care facilities. When LIBSS and ReSViNET were measured against each other, there was a strong correlation between the two scores (Pearson's >0.8). However, ReSViNET described fewer infants (n = 16, 16%) as having severe disease compared to LIBSS (n = 33, 33%) (Table 2). Therefore a larger study to identify appropriate cut-off points, along with responsiveness, would be an important piece of work to undertake.

## Conclusion

The findings of this study are important to the Rwandan health system where facilities have limited resources, and the decision to admit and escalate care must be carefully considered. This early data demonstrate that these two scores have the potential to be used in conjunction with clinical reasoning to identify infants at increased risk of clinical deterioration and allow timely admission, treatment escalation and therefore support resource allocation in Rwanda.

## Supporting information

**S1 File. TRIPOD checklist for ReSVinet LIBSS field testing.**
(DOCX)

**S2 File.**
(DOCX)

**S3 File.**
(DOCX)

**S4 File.**
(DOCX)

## Acknowledgments

Pediatric residents and nurses who collected data. Dr Federico Martinón-Torres and Dr Antonio Justicia of the ReSViNET team and Dr Paul McNamara of LIBSS for reviewing our manuscript.

## Author Contributions

**Conceptualization:** Boniface Hakizimana, Edgar Kalimba, Gemma Saint, Clare van Miert, Peter Thomas Cartledge.

**Data curation:** Boniface Hakizimana, Peter Thomas Cartledge.

**Formal analysis:** Boniface Hakizimana, Gemma Saint, Clare van Miert, Peter Thomas Cartledge.

**Investigation:** Boniface Hakizimana, Clare van Miert.

**Methodology:** Boniface Hakizimana, Edgar Kalimba, Augustin Ndatinya, Gemma Saint, Clare van Miert, Peter Thomas Cartledge.

**Project administration:** Boniface Hakizimana.

**Supervision:** Edgar Kalimba, Augustin Ndatinya, Gemma Saint, Clare van Miert, Peter Thomas Cartledge.

**Validation:** Boniface Hakizimana.

**Writing – original draft:** Boniface Hakizimana, Gemma Saint, Peter Thomas Cartledge.

**Writing – review & editing:** Edgar Kalimba, Augustin Ndatinya, Gemma Saint, Clare van Miert, Peter Thomas Cartledge.

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
