## [Decision Letter · Decision Letter 0]

2 Sep 2020

PONE-D-20-20016

Field testing two existing standardised respiratory severity scores (LIBSS and ReSViNET) in infants presenting with acute respiratory illness to tertiary hospitals in Rwanda – a validation and inter-rater reliability study

PLOS ONE

Dear Dr. Cartledge,

Thank you for submitting your manuscript to PLOS ONE. After careful consideration, we feel that it has some merit but requires substantial revision to meet PLOS ONE’s publication criteria. Therefore, we invite you to submit a revised version of the manuscript that addresses all the points raised during the review process.

We look forward to receiving your revised manuscript.

Kind regards,

Brenda M. Morrow, PhD

Academic Editor

PLOS ONE

Journal Requirements:

Reviewers' comments:

Reviewer's Responses to Questions

**Comments to the Author**

1. Is the manuscript technically sound, and do the data support the conclusions?

Reviewer #1: Partly

Reviewer #2: Partly

2. Has the statistical analysis been performed appropriately and rigorously? 

Reviewer #1: No

Reviewer #2: No

3. Have the authors made all data underlying the findings in their manuscript fully available?

Reviewer #1: Yes

Reviewer #2: No

4. Is the manuscript presented in an intelligible fashion and written in standard English?

Reviewer #1: No

Reviewer #2: Yes

5. Review Comments to the Author

Reviewer #1: Overall comments

The paper could benefit from English language editing. It is hard to follow and there are a number of grammatical errors. The main objective of the study is to assess two scores of respiratory distress from HIC one exclusively for bronchiolitis in infants. There is no clear motivation as to why the authors chose not to compare with other tools from LMIC settings taking into account high levels of malnutrition in their population. The patient selection and the motivation for item selection is not clearly articulated. There is no clear impression given to the context of the study. The inclusion and exclusion criteria and limited and the methodology is lacking in defining a number of variables that have been discussed in the document. There is no explanation why some sites failed to recruit any participants. There major limitation is the lack of uniformity of disease classification, 13% of study sample mentioned as "other". The representation of the data requires major revisions and the discussion is limited in its academic discussion of the findings and does not seek to address the primary objective and secondary objective.

Reviewer #2: This is an important research question that is of paramount importance in the care of children in low and middle income countries. There are a few fundamental issues that the authors need to consider in the methods and results section that I have pointed out. I have attached a detailed document with these comments...

6. PLOS authors have the option to publish the peer review history of their article (what does this mean?). If published, this will include your full peer review and any attached files.

Reviewer #1: No

Reviewer #2: No

---

## [Author Response · Author response to Decision Letter 0]

29 Jan 2021

Editorial team comments

We noticed minor instances of text overlap with the following previous publication(s), which need to be addressed: 

(1) https://academic.oup.com/tropej/article-abstract/66/2/234/5570307?redirectedFrom=fulltext

The text that needs to be addressed involves the Introduction section (lines 90-93).

 -Response: Thanks for pointing this out. We have put in alphabetical order of country and add our own paper reference.

Please amend either the title on the online submission form (via Edit Submission) or the title in the manuscript so that they are identical.

 -Response: Done

Comments to the Author 

Reviewer #1: Overall comments 

The paper could benefit from English language editing. It is hard to follow and there are a number of grammatical errors. 

 -Response: We are native English speakers and have used Grammarly to finish off the language. I’m not sure what else we can do, without having knowledge of the specific sections that are difficult to understand.

The main objective of the study is to assess two scores of respiratory distress from HIC one exclusively for bronchiolitis in infants. There is no clear motivation as to why the authors chose not to compare with other tools from LMIC settings taking into account high levels of malnutrition in their population. 

 -Response: We have moved around the introduction, which we had hoped explained their use. We have and created a new paragraph after the objectives to give an exaplanation of the reasoning for their use. This paragraph can be moved to the methods section if it is felt to be more appropriate.

The patient selection and the motivation for item selection is not clearly articulated. 

 -Response: Regarding patient selection, it was opportunistic based on the case-definition of “respiratory distress”. We have added this below the inclusion criteria under “sampling”. Regarding the item selection, we were limited to the items in the LIBSS and ReSViNET which we didn’t modify.

There is no clear impression given to the context of the study. 

 -Response: We are not sure what the reviewer would want in terms of context that is not described in the introduction?

The inclusion and exclusion criteria are limited 

 -Response: we’ve reworded this section, but these were the criteria we used, so there really isn’t much else to add.

The methodology is lacking in defining a number of variables that have been discussed in the document. 

 -Response: all the variables are described in the results and the questionnaires are available at the DOI. Therefore we didn’t want to add repetition.

There is no explanation why some sites failed to recruit any participants. 

 -Response: We’ve added this into the limitations.

There major limitation is the lack of uniformity of disease classification, 13% of study sample mentioned as "other". 

 -Response: we have split the other down to include the viral induced wheezing and URTI

The representation of the data requires major revisions and the discussion is limited in its academic discussion of the findings and does not seek to address the primary objective and secondary objective. 

 -Response: Each of the objectives is described in the discussion with a subheading. We would have welcomed the opportunity to go deeper into discussion, however to maintain a balanced and appropriate word count we have given a description that describes the findings in terms of the study population and a brief synopsis of it’s comparison with other studies. We have made general changes to the discussion and hopefully this is more satisfactory to the reviwere.

REVIEWER 1 COMMENTS (with responses)

Abstract: I would suggest that the ‘introduction’ contain some sort of rationale for the study

 -Response: we have added an additional sentence, but fear that the abstract word length may now be too long

Introduction: Line 89. Please check if this is indeed LIC or if it is supposed to be LMIC’s. Please note that these mean different things. For example, while South and Malawi are both LMICs, South Africa is Upper-Middle income, while Malawi is Low income…South Africa and India are not LIC

 -Response: Yes, this is an oversight of the terminology. We have amended this and reworded the paragraph. 

Materials And Methods

Line 131. So, to be clear, the children were selected on the basis of having respiratory distress. This would mean that the bases on which they were selected overlapped with the scoring system that will access the severity of their illness, right? This is important later given what the limitations are: the sample taken may not be appropriate to tell us how the tools will fare in detecting severe disease in unselected children (and in primary care levels, children arrive unselected).

 -Response: yes, this is a very valid point. We have added a section with the “validity” commentary in the discussion, rather then in the limitations 

Table 1. It may be useful to add columns that indicate the ranges of points for each item in each score. 

 -Response: we have come to realise that Table 1 isn’t that helpful, especially as many readers aren’t used to a Table in an introduction. We have therefore removed it. The scoring is all available in the tools which are available at the DOI.

Line 199. I have struggled to find these odd ratios in the paper. Did I miss them? Line 200. It is not clear as to what variables the model was adjusted for. What were these confounders? (I will return to this in the results section).

 -Response: Sorry, this was a remnant text from the PIs MMed dissertation. We removed this analysis from the manuscript due to the length and number of tables already presented. We have therefore deleted this from the methodlogy now.

 -Response: 

Lines 207-209. I am not entirely sure that doing correlation of two scores that have such a large overlap of components makes any sense at all…. In fact one does not need to collect data to figure out that they will almost be guaranteed to be correlated (as long as apnoea means the same thing in both scores, the fact that scores are allocated slightly differently should have little impact on correlation). In fact the correlation might potentially be seen as almost colinear from basic principles. I can therefore accept correlation of either score to the same but independent outcome, but not to each other… I hope my discomfort with this analysis makes sense (ReSVinet=70% of LIBBS’s components while 50% is the reverse) .

 -Response: We have added the following text to the statistics section of the methodology “The two scores use 12 different parameters. Five parameters are shared by the two tools, namely: Apnea; Feeding intolerance; General condition; Increased work of breathing; and respiratory rate. ReSViNET uses two additional parameters (Fever and medical intervention), and LIBSS employs five unique parameters (Appearance, Capillary refil time, heart rate, oxygen requirement and urine output). Therefore,…”

Would it not make sense to compare the severity of each score (a categorical analysis) to clinical severity as indicated in lines 193-194 and see how each ‘agree’ with this??

 -Response: we did a criterion validity for these surrogate markers of severity (see Table 4 in new version)

Results:

NB. As the study is about the two scoring methods, I expected to see a summary of the scores generated from these two, then see how they perform against clinical severity as mentioned above

 -Response: Is this related to the order we presented the results? We did think about this ourselves when writing the manuscript. We put the treatment quite early as we felt it provided more context on the practice of professionals at the study site. We have moved it down.

Lines 219-221: 107-8=99, however the data used in the rest of the paper as well as Fig 1 suggest that 7 (and not 8) were excluded.

 -Response: yes, sorry, this is an error, can’t believe we missed that. Sorry.

Line 228 – SD used for the first time here. Suggest writing it in full at first mention. 

 -Response: amended

By the way, I would have to confess that it is rather unusual for a number of the continuous variables reported here to be ‘normally’ distributed enough to be acceptably summarised with means and SDs. 

 -Response: amended. We’ve also added into the statistical analysis on how we categorized length of stay using the median.

Were all these data tested for normality (either formally or graphically). 

 -Response: we reviewed them graphically

Specifically with age, my suggestion would be to summarise this using months rather than days. It is just easier for the reader to immediately comprehend the age of the cohort that way. Hardly do people measure children’s age as days outside the neonatal period.

 -Response: amended, added months into the body of the text

NB. While, I really appreciate that for this cohort of 100 children, as long as we are referring to proportions of the total, the number will always equal the percentage, it is otherwise good practice to always report both number and percentage [n(%)] in the text. When part of the report includes subgroup proportions, not having numbers to which the percentages refer gets very confusing.

 -Response: amended.

Table 2. 

Table includes ‘social group’ which is undefined 

 -Response: amended, footnote added

‘Vaccination’ what does No and Yes mean?

 -Response: amended, sorry

Maternal education: What does high and low mean??

 -Response: amended, footnote added

‘Median length of stay’ but result indicate some sort of standard deviation??

 -Response: amended

Table 3. By the way, the use of a Kappa statistic is coming here for the first time being absent in methods.

 -Response: Sorry, this column should have been removed from the table. It is not needed as the ICC is described in Table 4.

Line 248. I think it should read Table 4 not 3

 -Response: amended, now Table 6, see above

NB. When it comes to tables and figures, make sure all acronyms are explained in the figure/table legends. E.g. acronyms such as ‘ICC’ in figures..

 -Response: amended

Table 4. What is the difference between the ‘others’ of “Bronchiolitis and others” and ‘Other’?

 -Response: sorry, this is an error. The “and others” has been removed”

Table 5. Already alluded to in Methods, I am not sure what is adjusted for here

 -Response: see above. The adjusted odds ratios were a separate analysis in the PIs dissertation. 

Table 5 - Would it be possible to show the data for each binary outcome?

 -Response: It would make very heavy tables, therefore we chose not to. Original data is available

Table 5 - Secondly, in the case of adjustment being done one would need to show both adjusted and unadjusted estimates

 -Response: see above. The adjusted odds ratios were a separate analysis in the PIs dissertation.

Table 5 - Length of hospital stay: I can see the others working as ‘binary’ in a Yes-No gold standard to generate 

 -Response: Yes, we realise now we never described how we did this. We’ve added it to the statistical analysis section.

Some things in the table legend don’t seem to belong to it: Cronbach, SPR

 -Response: amended

Discussion

My sense is that the section on treatment (Lines 300-314) is an unwarranted digression that important as it is has little to do with the theme of the paper. If there is a need for it, it really needs to be brief 

 -Response: We hadn’t initially put this in the “objectives” as the objectives statement was already quite long. We’ve added this back in. Also we have cut down this section in the discussion.

Lines 338-340. I can’t see anything in table 6 indicating the detrimental effect of apnoea to reliability. Maybe if there is anything of the sort the authors could highlight this in the text in the results section before discussing it here.

 -Response: We’ve added it to the text above the table (which is now labelled Table 4)

Lines 343-345; Could the authors not have gone around this issue by doing stratified analysis with just bronchiolitis to see if they find results similar to those in HIC?

 -Response: We didn’t do this.

Conclusion

Are the authors suggesting that the two scores yield similar results? It just seems to me that table 3 suggests that one score misses half of severe cases (or maybe one wrongly doubles the number of severe cases??) This is worrying and needs explanation, but can not be addressed unless the two scores were assessed against a single ‘gold’ standard as suggested previously rather than to each other. 

 -Response: We’ve added the following text to the “next steps” section: “When LIBSS and ReSViNET were measured against each other, there was a strong correlation between the two scores (Pearson's >0.8). However, ReSViNET described fewer infants (n=16, 16%) as having severe disease compared to LIBSS (n=33, 33%) (Table 2). Therefore a larger study to identify appropriate cut-off points, along with responsiveness, would be an important piece of work to undertake.”

REVIEWER 2 COMMENTS and Responses

The representation of the data requires major revisions and the discussion is limited in its academic discussion of the findings and does not seek to address the primary objective and secondary objective. 

 -Response: This is quite a broad feedback, therefore hopefully the points are addressed in the below amendments and responses.

The tables and figures are not comprehensively captioned and there are mistakes when the author makes references to some tables. There are two (Table 1s) in the text. 

 -Response: amended

The authors do not use the discussion to elaborate fully on their results they mention that findings are interesting and does try to elaborate, compare and make arguments for or against.

 -Response: As discussed above, we are limited by the number of words and didn’t want the balance of the paper to fall mainly on the discussion. Most readers, in our opinion, can make their own conclusions and opinions without us going through each result in detail.

References are made to a PhD thesis that is not available in print, textbooks where pages and chapters are not mentioned, references to studies that were apparently done in the UK but when looked up are somewhere else, conclusive remarks supported by comparison to studies with different methodologies (compares apples with oranges). 

 -Response: The entire PhD is available freely as a DOI. If the reference to the study “apparently” done in the UK was the Griffiths, and Riphagen article then I can confirm it was done in the UK. Of course, the reviewer is right that comparing UK with Rwanda is challenging, but that is the nature of our paper, using a tool developed in UK (LIBSS) for it’s appropriateness in Rwanda. 

The reporting of numbers is not the same everywhere at times uses 3 decimal points then uses 2, the results in abstract are not the same as those in the tables in results.

 -Response: Yes, this is challenging. For the numbers <1 we tended to use three decimal places. We’ve amended the abstract to use 2 decimal places for the Cis.

The introduction is fair but for the discussion, the author has not tried to put create a followable argument, there are a lot of headings with paragraphs that do not have an introduction, body and conclusion, there are just thoughts put under a heading. So the gist of the article is lost. 

 -Response: We have made a number of changes to the discussion, but without the word count being significantly increased I think it is unlikely that this will have been fully addressed to the reviewers requirement.

The paper argues for use of a tool developed in resource-rich but does not explain why? e.g. What's wrong with ones developed in LMICs? 

 -Response: Amended, as per previous peer-reviewer comments

Also if it will be used to use in different resp illnesses it's no longer a validation study but an adaptation or modified tool. or further development of the tool. If the study compared its use in bronchiolitis only it would validate the tool. I am not convinced about the validation argument.

 -Response: We were field testing it. We are aware that significantly more work will need to be done to fully validate it. As discussed in the “next steps” section.

Methodology

Bias: participant selection would be those with severe disease if no self-referral system- how can this be applicable to patients with mild or moderate disease

 -Response: We have added a new paragraph called “bias” to the discussion and have discussed this within that section 

How were the participants selected: this is not clear ??

 -Response: Amended, as per reviewer 1 comments

Other relevant exclusions??

 -Response: No other exclusions

Table 1; for ease of reading useful to have similar markers for both scores in same line so easy for reader to see which parameters are dissimilar. Also two tables named Table 1.

 -Response: We’ve now removed Table 1. And we’ve added the similar markers into the text

There is no explanation of what is meant by Social Group (Ubedehe) High (3 & 4),Low (1 & 2), what levels of schooling etc for parents. 

 -Response: added into footnote of the tables

What vaccination was asked about, what is the EPI program in Rwanda?

 -Response: Sorry, it was just whether they had received vaccines or not. We have added this to the table

Results

The final number stated is 100 but only 99 were analyzed as there were 107 and 8 exclusions? “from eight infants could not be used due to data collection being incomplete or inadequate”

 -Response: This was an error. We’ve amended it. There were 100

Table 1 missing after characteristics no explanation of what data is presented

 -Response: This “Table 1” has been removed and it’s contents moved into the body of the text.

Table 1: 28 required HDU and ICU but 13 seem to have received interventions in HDU or PICU

 -Response: We aren’t exactly sure what the point is that the reviewer is making. Sorry.

Lack of consistency with decimal places, confusing results and explanation of key in tables and figures

 -Response: we have used three decimal places for numbers with a value less than 1. Except for confidence intervals, for which we used 2 decimal places. We have added additional information to the keys for tables and figures, as per reviewer 1.

Discussion 

There is no logical flow to the discussion and this fails to give a detailed academic discussion on the results of the study and the context.

 -Response: as above

Conclusion: considering the study size, variable disease profiles of children I think the conclusion is slightly over-reaching in the abstract, the conclusion in the main document more modest. 

 -Response: amended

---

## [Decision Letter · Decision Letter 1]

8 Oct 2021

Field testing two existing, standardized respiratory severity scores (LIBSS and ReSViNET) in infants presenting with acute respiratory illness to tertiary hospitals in Rwanda – a validation and inter-rater reliability study

PONE-D-20-20016R1

Dear Dr. Cartledge,

We’re pleased to inform you that your manuscript has been judged scientifically suitable for publication and will be formally accepted for publication once it meets all outstanding technical requirements.

Kind regards,

Jamie Males

Staff Editor

PLOS ONE

Additional Editor Comments (optional):

We noticed minor instances of text overlap with the following previous publication(s), which need to be addressed:

(1) https://academic.oup.com/tropej/article-abstract/66/2/234/5570307?redirectedFrom=fulltext

The text that needs to be addressed involves the Introduction section (lines 90-93).

In your revision please ensure you cite all your sources (including your own works), and quote or rephrase any duplicated text outside the methods section. Further consideration is dependent on these concerns being addressed.

Reviewers' comments:

Reviewer's Responses to Questions

**Comments to the Author**

1. If the authors have adequately addressed your comments raised in a previous round of review and you feel that this manuscript is now acceptable for publication, you may indicate that here to bypass the “Comments to the Author” section, enter your conflict of interest statement in the “Confidential to Editor” section, and submit your "Accept" recommendation.

Reviewer #1: All comments have been addressed

2. Is the manuscript technically sound, and do the data support the conclusions?

Reviewer #1: Yes

3. Has the statistical analysis been performed appropriately and rigorously? 

Reviewer #1: Yes

4. Have the authors made all data underlying the findings in their manuscript fully available?

Reviewer #1: Yes

5. Is the manuscript presented in an intelligible fashion and written in standard English?

Reviewer #1: Yes

6. Review Comments to the Author

Reviewer #1: (No Response)

7. PLOS authors have the option to publish the peer review history of their article (what does this mean?). If published, this will include your full peer review and any attached files.

Reviewer #1: No

---

## [Editor Report · Acceptance letter]

26 Oct 2021

PONE-D-20-20016R1 

Field testing two existing, standardized respiratory severity scores (LIBSS and ReSViNET) in infants presenting with acute respiratory illness to tertiary hospitals in Rwanda – a validation and inter-rater reliability study 

Dear Dr. Cartledge:

I'm pleased to inform you that your manuscript has been deemed suitable for publication in PLOS ONE. Congratulations! Your manuscript is now with our production department. 

Kind regards, 

on behalf of

Dr Jamie Males 

Staff Editor

PLOS ONE